# Functional impact of subunit composition and compensation on *Drosophila melanogaster* nicotinic receptors–targets of neonicotinoids

Yuma Komori[1☺], Koichi Takayama[1☺], Naoki Okamoto[2☺], Masaki Kamiya[1], Wataru Koizumi[1], Makoto Ihara[1], Daitaro Misawa[3], Kotaro Kamiya[3], Yuto Yoshinari[2¤], Kazuki Seike[4], Shu Kondo[5,6], Hiromu Tanimoto[7], Ryusuke Niwa[2], David B. Sattelle[8], Kazuhiko Matsuda[1,2,9]*

1 Department of Applied Biological Chemistry, Faculty of Agriculture, Kindai University, Nara, Japan, 2 Life Science Center for Survival Dynamics, Tsukuba Advanced Research Alliance (TARA), University of Tsukuba, Tsukuba, Japan, 3 SyntheticGestalt, KK, Tokyo, Japan, 4 Degree Programs in Life and Earth Sciences, Graduate School of Science and Technology, University of Tsukuba, Tsukuba, Japan, 5 Department of Biological Science and Technology, Faculty of Advanced Engineering, Tokyo University of Science, Tokyo, Japan, 6 Invertebrate Genetics Laboratory, National Institute of Genetics, Shizuoka, Japan, 7 Graduate School of Life Sciences, Tohoku University, Miyagi, Japan, 8 Centre for Respiratory Biology, Division of Medicine, University College London, London, United Kingdom, 9 Agricultural Technology and Innovation Research Institute, Kindai University, Nara, Japan

☺ These authors contributed equally to this work.
¤ Current address: Institute for Molecular and Cellular Regulation, Gunma University, Maebashi, Japan.
* kmatsuda@nara.kindai.ac.jp

**Data Availability Statement:** The data used for analysis in this study are available from the Dryad (https://doi.org/10.5061/dryad.qz612jmk5).

## Abstract

Neonicotinoid insecticides target insect nicotinic acetylcholine receptors (nAChRs) and their adverse effects on non-target insects are of serious concern. We recently found that cofactor TMX3 enables robust functional expression of insect nAChRs in *Xenopus laevis* oocytes and showed that neonicotinoids (imidacloprid, thiacloprid, and clothianidin) exhibited agonist actions on some nAChRs of the fruit fly (*Drosophila melanogaster*), honeybee (*Apis mellifera*) and bumblebee (*Bombus terrestris*) with more potent actions on the pollinator nAChRs. However, other subunits from the nAChR family remain to be explored. We show that the Dα3 subunit co-exists with Dα1, Dα2, Dβ1, and Dβ2 subunits in the same neurons of adult *D. melanogaster*, thereby expanding the possible nAChR subtypes in these cells alone from 4 to 12. The presence of Dα1 and Dα2 subunits reduced the affinity of imidacloprid, thiacloprid, and clothianidin for nAChRs expressed in *Xenopus laevis* oocytes, whereas the Dα3 subunit enhanced it. RNAi targeting *Dα1*, *Dα2* or *Dα3* in adults reduced expression of targeted subunits but commonly enhanced *Dβ3* expression. Also, *Dα1* RNAi enhanced *Dα7* expression, *Dα2* RNAi reduced *Dα1*, *Dα6*, and *Dα7* expression and *Dα3* RNAi reduced *Dα1* expression while enhancing *Dα2* expression, respectively. In most cases, RNAi treatment of either *Dα1* or *Dα2* reduced neonicotinoid toxicity in larvae, but *Dα2* RNAi enhanced neonicotinoid sensitivity in adults reflecting the affinity-reducing effect of Dα2. Substituting each of Dα1, Dα2, and Dα3 subunits by Dα4 or Dβ3 subunit mostly increased neonicotinoid affinity and reduced efficacy. These results are important because they indicate that neonicotinoid actions involve the integrated activity of multiple

**Funding:** This study was supported by KAKENHI (Grant-in-Aid for Scientific Research) from the Japan Society for the Promotion of Science (grant number 21H04718 (KM and RN), 22H02350 (MI),26250001 (HT), and 17H01378 (HT)); and the Cooperative Research Project Program of Life Science Center for Survival Dynamics, Tsukuba Advanced Research Alliance (TARA Center), University of Tsukuba, Japan (grant number 202113 and 202217 (KM)).The funders had no role in study design, data collection and analysis, decision to publish, or preparation of the manuscript.

**Competing interests:** The authors have declared that no competing interests exist.

nAChR subunit combinations and counsel caution in interpreting neonicotinoid actions simply in terms of toxicity.

## Author summary

In this paper, we show that the *Drosophila melanogaster* nicotinic acetylcholine receptor (nAChR) Dα3 subunit is co-expressed in ejaculatory duct neurons with Dα1, Dα2, Dβ1, and Dβ2 subunits. All 5 subunits combine to form 12 functional nAChRs in *Xenopus laevis* oocytes. The functional expression of 18 nAChRs generated from combinations of subunits Dα1−4 and Dβ1−3 are also reported. Dα1 and Dα2 reduced the affinity of *D. melanogaster* heteromeric nAChRs for imidacloprid, thiacloprid, and clothianidin, whereas Dα3 enhanced it. RNAi of *Dα1*, *Dα2* or *Dα3* in adult flies reduced expression of the targeted subunits but commonly enhanced *Dβ3* expression; other subunits were also affected in some cases. RNAi targeting either *Dα1* or *Dα2* reduced neonicotinoid toxicity in larvae but targeting *Dα2* led to hyper-neonicotinoid sensitivity in adults consistent with the affinity-reducing effect on neonicotinoids of Dα2. Since RNAi induced subunit compensation was detected, each of Dα1, Dα2, and Dα3 subunits was substituted by Dα4 or Dβ3 subunit. Such subunit compensation mostly increased neonicotinoid affinity and reduced efficacy, impairing the climbing ability of the flies. These results are important because they indicate that neonicotinoid action and toxicity involve the integrated actions of multiple nAChR subunit combinations and counsel caution in interpreting neonicotinoid actions in terms of reduced toxicity.

## Introduction

The nicotinic acetylcholine receptors (nAChRs) are cys-loop ligand-gated cation channels playing a pivotal role in fast cholinergic neurotransmission [1]. In mammals, nAChRs underlie memory [2], learning [2], circadian rhythm [3] and immune responses [4] as well as locomotion [5] and hearing [6, 7]. In insects, nAChRs are involved primarily in afferent synaptic transmission [8]. Roles for insect nAChRs include escape responses [9], circadian rhythm [10,11] and regulation of the mating-induced germline stem cell growth [12]. Hence, several classes of synthetic and natural-product based insecticides target nAChRs [13–18].

Neonicotinoid insecticides targeting insect nAChRs are effective, broad-spectrum insecticides [16,19–23]. Following the discovery of the nitromethylene heterocyclic compound nithiazine, this initial lead compound was modified extensively resulting in 3 generations of commercial neonicotinoids [15]. They exhibit higher affinity for insect over vertebrate nAChRs, thereby resulting in selective toxicity to insects [24,25]. Their high systemic activity in plants has permitted seed treatment which has accelerated their deployment for crop protection. However, potential adverse effects on pollinators such as honeybees, bumblebees and solitary bees are a concern [16,20,26,27], even though reduced numbers of pollinators and other non-target insects are not simply due to the effects of neonicotinoids [16,26]. Adverse effects on aquatic invertebrates and birds are also reported [28,29]. It is vital to understand in detail the mechanism of insect nAChR-neonicotinoid interactions but until recently that was precluded by the difficulty of heterologously expressing robust insect nAChRs.

The recent finding that the transmembrane thioredoxin-related protein 3 (TMX3) enables robust functional expression of insect nAChRs in *X. laevis* oocytes [20, 30, 31] permitted a

demonstration that honeybee (*A. mellifera*) and bumblebee (*B. terrestris*) α1/α8/β1 and α1/α2/α8/β1 nAChRs were more sensitive to neonicotinoids than the fruit fly (*D. melanogaster*) Dα1/Dβ1, Dα1/Dα2/Dβ1, Dα1/Dβ1/Dβ2, and Dα1/Dα2/Dβ1/Dβ2 nAChRs. Thiacloprid and clothianidin modulated honeybee and bumblebee nAChRs at picomolar concentrations, lower than those commonly encountered in treated fields [30]. However, the extent to which other subunits participate in the formation of native nAChRs and how that affects neonicotinoid actions is not known.

To begin to address this shortfall in understanding, we first showed that the *D. melanogaster* Dα3 subunit co-exists with the Dα1, Dα2, Dβ1, and Dβ2 subunits in ejaculatory duct neurons of adult *D. melanogaster*. Co-expression in *X. laevis* with the Dα3 subunit increased possible nAChR subtypes from 4 to 12. All the 12 recombinant fruit fly nAChRs will be explored and their sensitivity to the transmitter acetylcholine (ACh), neonicotinoids (imidacloprid, thiacloprid, and clothianidin) and α-bungarotoxin compared. The impact of the Dα1, Dα2, and Dα3 subunits on recombinant nAChR receptor affinity is addressed. The impact of RNAi targeting particular subunits on toxicity to larvae and adult toxicity and behaviour is investigated. Finally, we studied the effects of replacing one of the α subunits in the Dα1/Dα2/Dα3/Dβ1/Dβ2 nAChRs by either the Dα4 or the Dβ3 subunit on the neonicotinoid actions to address whether such subunit compensation further expands the diversity of neonicotinoid actions in insects.

## Results and discussion

We first examined whether the Dα3 subunit is co-expressed with Dα1, Dα2, Dβ1, and Dβ2 subunits, previously shown to be present in ejaculatory neurons of *D. melanogaster* [30], and if so, how such co-expression influences actions of neonicotinoids *in vitro* and *in vivo*. To analyse the expression of *Dα3*, we stained male ejaculatory neurons with a tyrosine decarboxylase 2 targeting antibody (anti-Tdc2) in the animals expressing *GFP* under control of *Dα3-T2A-Gal4* and found that *Dα3* is indeed expressed in the male ejaculatory neurons (**Fig 1A**). Another recent study has shown similar findings for the oviduct neurons in female fruit flies [12], suggesting that Dα1, Dα2, Dα3, Dβ1, and Dβ2 subunits can potentially generate diverse heteromeric nAChRs in male and female adult neurons involved in reproductive functions.

Given their co-localisation in certain neurons, we investigated in terms of responses to bath-applied 100 μM ACh how many kinds of robust, functional nAChRs the five subunits can reconstitute in *X. laevis* oocytes when co-expressed with co-factors DmRIC-3, DmUNC-50, and DmTMX3 [30]. We found that by co-expressing the Dα3 subunit, 8 more robust nAChRs (Dα3/Dβ1, Dα1/Dα3/Dβ1, Dα2/Dα3/Dβ1, Dα1/Dα2/Dα3/Dβ1, Dα3/Dβ1/Dβ2, Dα1/Dα3/Dβ1/Dβ2, Dα2/Dα3/Dβ1/Dβ2, and Dα1/Dα2/Dα3/Dβ1/Dβ2 nAChRs) were generated in addition to previously described Dα1/Dβ1, Dα1/Dα2/Dβ1, Dα1/Dβ1/Dβ2, and Dα1/Dα2/Dβ1/Dβ2 nAChRs (**Fig 1B**). We then determined concentration-response relationships for ACh on the 12 nAChRs (**Fig A in S1 Text and Table 1**). Replacing either the Dα1 or Dα2 subunit by the Dα3 subunit led to increased current amplitude of the response to 100 μM ACh. For example, the ACh response amplitudes of the Dα3/Dβ1 and Dα1/Dα3/Dβ1 nAChRs were 58.2 and 7.1-fold larger than those of Dα1/Dβ1 and Dα1/Dα2/Dβ1 nAChRs, respectively (ANOVA, $P < 0.05$, n = 10, **Fig 1B**). The affinity in terms of $pEC_{50}$ for ACh varied from 4.14 to 6.14 depending on subunit combinations (**Fig 2A, Table 1, and Table A in S1 Text**), indicative of 12 distinct nAChRs. Notably, the Dα1/Dα2/Dα3/Dβ1/Dβ2 nAChR is the first functional recombinant insect nAChR consisting of five different subunits. It is also the first time that all possible subunit combinations for a single insect neuron have been reported.

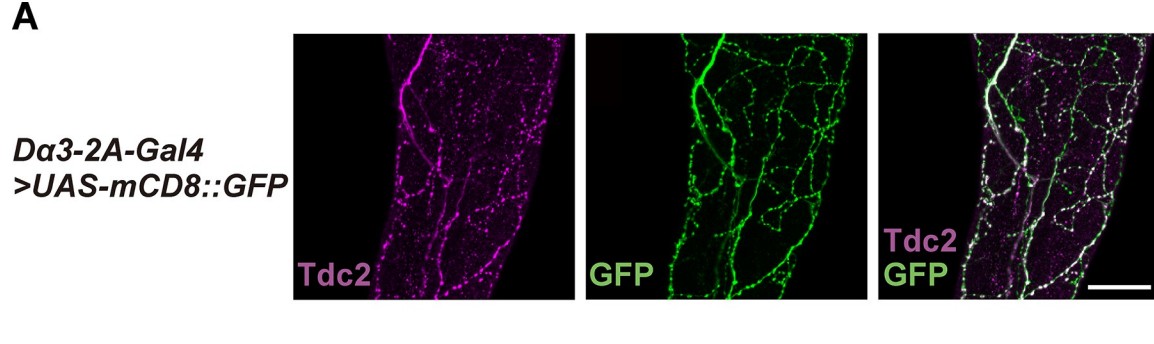

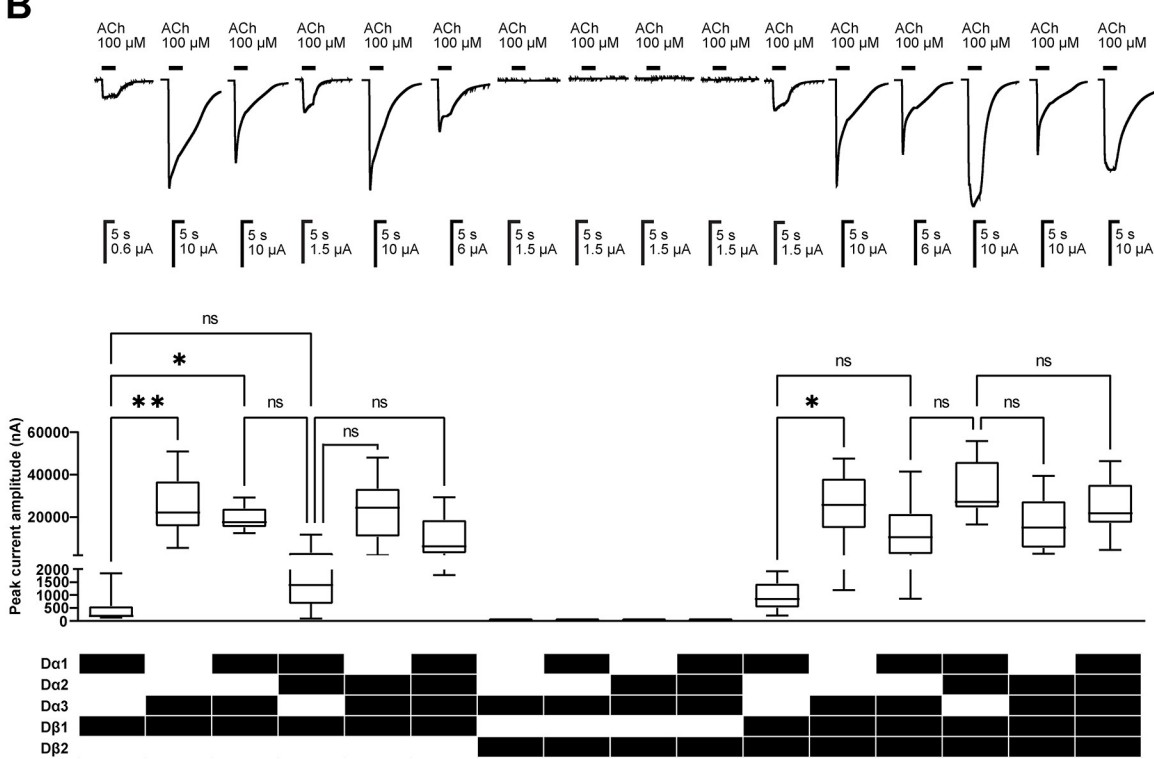

**Fig 1. Expression of *Dα3* in male ejaculatory duct neurons of *D. melanogaster* and characteristics of nAChRs expressed in *X. laevis* oocytes.** (A) *Dα3* is expressed in the ejaculatory ducts of male flies. Male ejaculatory ducts from *Dα3-2A-Gal4>UAS-mCD8::GFP* were immunostained for Tdc2 (magenta) and GFP (green). If both signals are overlapped, merged signals are shown in white. Scale bar: 25 μm. The Dα3 subunit is expressed in the neurons where Dα1, Dα2, Dβ1, and Dβ2 subunits are also expressed [30]. (B) Current responses to 100 μM ACh of nAChRs reconstructed with Dα1, Dα2, Dα3, Dβ1, and Dβ2 subunits in *X. laevis* oocytes. Each box plots represents 75 and 25% percentiles of data and horizonal line in each box indicates the median of data (n = 10). Whiskers indicate the range of data. The current amplitude of the ACh-induced response was compared by Kruskal-Wallis tests (*, $P < 0.05$; **, $P < 0.01$). ns: not significant.

α-Bungarotoxin (α-BTX), a peptide toxin known to block certain insect nAChRs [32], was tested on *D. melanogaster* nAChRs expressed in *X. laevis* oocytes. We found that 100 nM α-BTX effectively blocked the ACh-induced responses of nAChRs containing the Dα1 subunit (**Fig 2B, and Fig B in S1 Text**). In contrast, α-BTX was ineffective on nAChRs lacking the Dα1 subunit. Notably, α-BTX showed a minimal blocking effect on the Dα3/Dβ1, Dα2/Dα3/ Dβ1, and Dα3/Dβ1/Dβ2 nAChRs (**Fig 2B and Fig B in S1 Text**). These findings accord with an earlier observation that Dα1/chicken β2 nAChR was sensitive whereas Dα2/chicken β2

**Table 1. Agonist actions of ACh and neonicotinoids on *D. melanogaster* nAChRs expressed in *X. laevis* oocytes[\*].**

| Subunits | ACh | Imidacloprid | | Thiacloprid | | Clothianidin | |
|---|---|---|---|---|---|---|---|
| | $pEC_{50}$ | $pEC_{50}$ | $I_{max}$ | $pEC_{50}$ | $I_{max}$ | $pEC_{50}$ | $I_{max}$ |
| Dα1/Dβ1 | 5.24 ± 0.04 | 7.16 ± 0.15 | 0.122 ± 0.007 | 7.79 ± 0.21 | 0.091 ± 0.009 | 7.13 ± 0.10 | 0.508 ± 0.015 |
| Dα3/Dβ1 | 6.14 ± 0.06 | 8.32 ± 0.21 | 0.032 ± 0.003 | 7.80 ± 0.08 | 0.016 ± 0.001 | 7.54 ± 0.04 | 0.691 ± 0.013 |
| Dα1/Dα3/Dβ1 | 5.85 ± 0.03 | 7.58 ± 0.23 | 0.170 ± 0.020 | 8.07 ± 0.20 | 0.047 ± 0.006 | 7.20 ± 0.09 | 0.627 ± 0.025 |
| Dα1/Dα2/Dβ1 | 4.14 ± 0.05 | 6.71 ± 0.14 | 0.031 ± 0.002 | 6.92 ± 0.08 | 0.025 ± 0.001 | 5.07 ± 0.11 | 0.699 ± 0.034 |
| Dα2/Dα3/Dβ1 | 5.45 ± 0.07 | 6.88 ± 0.29 | 0.101 ± 0.016 | 6.92 ± 0.19 | 0.045 ± 0.006 | 6.81 ± 0.08 | 0.882 ± 0.076 |
| Dα1/Dα2/Dα3/Dβ1 | 5.43 ± 0.05 | 7.08 ± 0.15 | 0.094 ± 0.007 | 7.73 ± 0.19 | 0.047 ± 0.004 | 6.55 ± 0.07 | 0.817 ± 0.024 |
| Dα1/Dβ1/Dβ2 | 4.75 ± 0.04 | 6.99 ± 0.18 | 0.076 ± 0.006 | 7.86 ± 0.10 | 0.043 ± 0.002 | 6.64 ± 0.10 | 0.360 ± 0.016 |
| Dα3/Dβ1/Dβ2 | 6.19 ± 0.07 | 8.28 ± 0.09 | 0.039 ± 0.002 | 8.32 ± 0.21 | 0.022 ± 0.003 | 7.46 ± 0.08 | 0.492 ± 0.016 |
| Dα1/Dα3/Dβ1/Dβ2 | 6.43 ± 0.02 | 7.52 ± 0.12 | 0.330 ± 0.018 | 8.00 ± 0.25 | 0.107 ± 0.015 | 7.40 ± 0.07 | 0.684 ± 0.022 |
| Dα1/Dα2/Dβ1/Dβ2 | 4.92 ± 0.05 | 6.45 ± 0.21 | 0.165 ± 0.016 | 7.10 ± 0.30 | 0.074 ± 0.014 | 6.31 ± 0.10 | 0.800 ± 0.033 |
| Dα2/Dα3/Dβ1/Dβ2 | 5.99 ± 0.07 | 7.00 ± 0.29 | 0.125 ± 0.018 | 8.08 ± 0.33 | 0.017 ± 0.002 | 7.29 ± 0.07 | 0.634 ± 0.020 |
| Dα1/Dα2/Dα3/Dβ1/Dβ2 | 4.49 ± 0.04 | 6.47 ± 0.14 | 0.142 ± 0.011 | 7.18 ± 0.17 | 0.059 ± 0.006 | 6.59 ± 0.07 | 0.812 ± 0.019 |
| Dα2/Dα3/Dα4/Dβ1/Dβ2 | 5.49 ± 0.06 | 7.40 ± 0.20 | 0.027 ± 0.002 | 7.99 ± 0.20 | 0.018 ± 0.002 | 7.11 ± 0.12 | 0.912 ± 0.041 |
| Dα2/Dα3/Dβ1/Dβ2/Dβ3 | 6.01 ± 0.04 | 7.75 ± 0.31 | 0.021 ± 0.003 | 8.31 ± 0.17 | 0.010 ± 0.001 | 7.46 ± 0.06 | 0.695 ± 0.014 |
| Dα1/Dα3 Dα4/Dβ1/Dβ2 | 4.79 ± 0.04 | 7.30 ± 0.17 | 0.101 ± 0.008 | 7.96 ± 0.10 | 0.026 ± 0.001 | 6.94 ± 0.06 | 0.425 ± 0.008 |
| Dα1/Dα3/Dβ1/Dβ2/Dβ3 | 5.83 ± 0.02 | 7.72 ± 0.11 | 0.140 ± 0.008 | 8.63 ± 0.23 | 0.089 ± 0.009 | 7.60 ± 0.12 | 0.531 ± 0.026 |
| Dα1/Dα2/Dα4/Dβ1/Dβ2 | 4.64 ± 0.05 | 6.48 ± 0.28 | 0.172 ± 0.020 | 7.01 ± 0.19 | 0.066 ± 0.007 | 6.72 ± 0.09 | 1.001 ± 0.031 |
| Dα1/Dα2/Dβ1/Dβ2/Dβ3 | 4.42 ± 0.02 | 6.27 ± 0.21 | 0.080 ± 0.007 | 7.71 ± 0.43 | 0.023 ± 0.004 | 5.67 ± 0.11 | 0.974 ± 0.041 |

[\*]All data are shown as mean ± standard error (n = 5). For this study, we determined $pEC_{50}$ and $I_{max}$ values of the ligands for the Dα1/Dβ1, Dα1/Dα2/Dβ1, Dα1/Dα2/Dβ1/Dβ2 and Dα1/Dα2/Dβ1/Dβ2 nAChRs for which data were reported previously [30], because it was necessary to compare all the data determined using the same protocol (see Methods).

nAChR was resistant to this neurotoxin [33]. Diverse pharmacology in terms of the sensitivity to α-BTX confirms that 12 distinct, robust, and functional nAChRs results from combinatorial assembly from the five subunits.

We measured agonist activity of imidacloprid, thiacloprid, and clothianidin, in terms of their $pEC_{50}$ and $I_{max}$ values for 8 Dα3-containing nAChRs and analysed factors determining them (**Fig 3A, Fig C in S1 Text** and **Table 1**). The $pEC_{50}$ value for each neonicotinoid relies primarily on the subunit properties (**Fig 3B** and **Table 1**). For imidacloprid, the $EC_{50}$ value for the Dα3/Dβ1 nAChR was 14.4-fold lower than that for the Dα1Dβ1 nAChR. Similarly, substituting either Dα1 or Dα2 subunit by Dα3 subunit enhanced affinity of imidacloprid and clothianidin (Dα1/Dβ1/Dβ2 nAChR (for imidacloprid, $pEC_{50}$ = 6.99, $EC_{50}$ = 102 nM; for clothianidin, $pEC_{50}$ = 6.64, $EC_{50}$ = 229 nM) vs Dα3/Dβ1/Dβ2 nAChR (for imidacloprid, $pEC_{50}$ = 8.28, $EC_{50}$ = 5.25 nM; for clothianidin, $pEC_{50}$ = 7.46, $EC_{50}$ = 34.7 nM), **Fig 3B**, **Table 1**, and **Tables B and F in S1 Text** for ANOVA analysis). For thiacloprid, the affinity for the Dα1/Dα3/Dβ1 nAChR ($pEC_{50}$ = 8.07, $EC_{50}$ = 8.51 nM) was higher than that for the Dα1/Dα2/Dβ1 nAChR ($pEC_{50}$ = 6.92, $EC_{50}$ = 120 nM, **Fig 3B**, **Table 1**, and **Table D in S1 Text** for ANOVA analysis). Inversely, the Dα2 subunit reduced the affinity of neonicotinoids (imidacloprid and clothianidin, Dα2/Dα3/Dβ1 nAChR < Dα3/Dβ1nAChR; thiacloprid, Dα2/Dα3/Dβ1 nAChR < Dα1/Dα3/Dβ1 nAChR, **Tables B, D, and F in S1 Text** for ANOVA analysis). Compound properties also contribute to determining the affinity as indicated by the highest $pEC_{50}$ values of thiacloprid for most of the nAChRs (**Fig 3A and 3B**, **Table 1**, **Tables B**, **D**, **and F in S1 Text** for ANOVA analysis). Such compound factors were more evident in the $I_{max}$ values (**Fig 3C**, **Table 1**, **and Tables C**, **E**, **and G in S1 Text** for ANOVA analysis). For all the nAChRs tested, the order of $I_{max}$ was clothianidin > imidacloprid > thiacloprid, similar to the

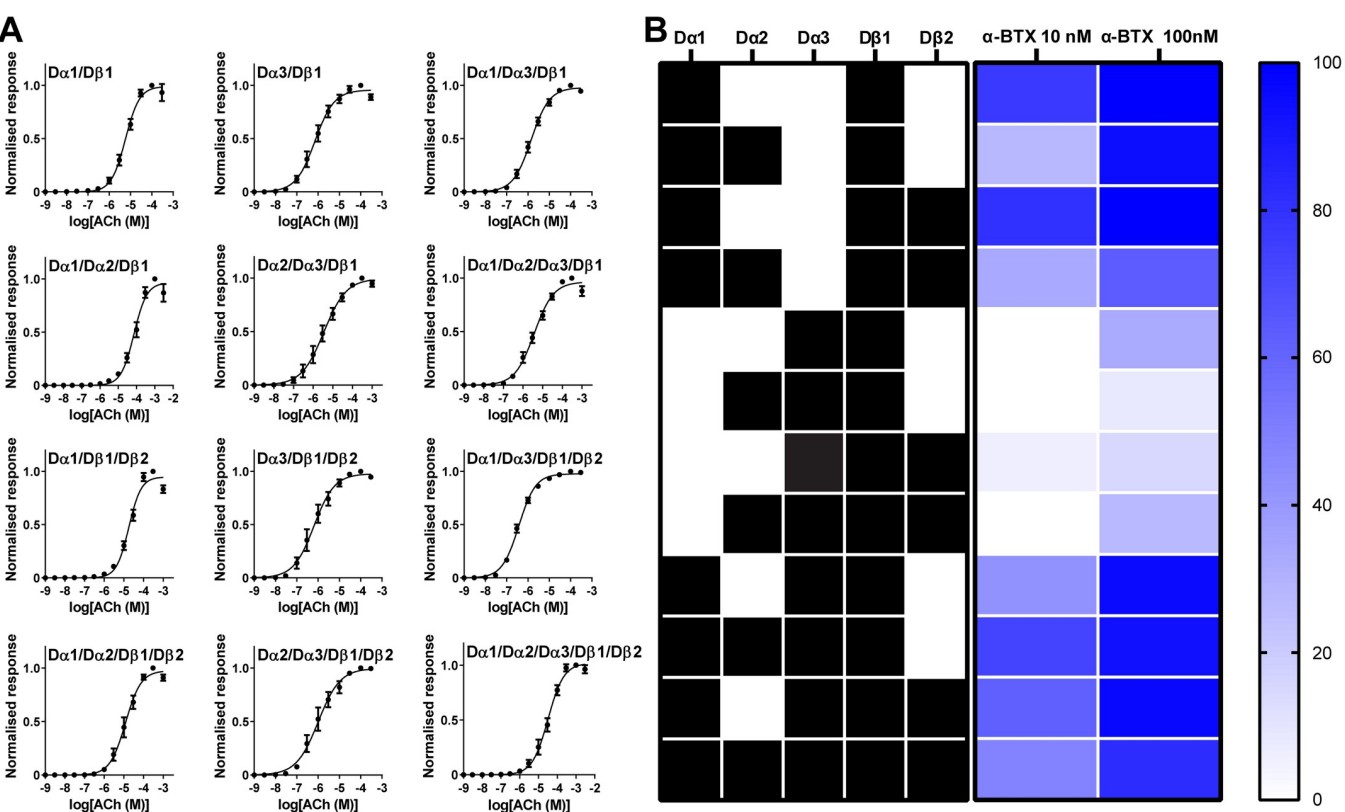

**Fig 2. Concentrations-response relationships of ACh for *D. melanogaster* nAChRs expressed in *X. laevis* oocytes and actions of α-BTX on the ACh-induced responses of nAChRs.** (**A**) Concentration-response curves of ACh and (**B**) heatmap representing α-BTX for the 12 nAChRs expressed in *X. laevis* oocytes. In (**A**), each data plot indicates the mean ± standard error (n = 5). In (**B**), high to low α-BTX sensitivity is shown in blue and white, respectively. The Dα1 subunit underpins the α-BTX sensitivity of the nAChRs tested. See **Fig B in S1 Text** for the ACh-induced currents measured in the absence and presence of α-BTX (**Fig Ba in S1 Text**) and bar graph representations of the effects of α-BTX (**Fig Bb in S1 Text**).

efficacy order observed in the fruit fly neurons [34], which supports the utility of using the *X. laevis* oocytes to express nAChRs for the evaluation of neonicotinoid actions in insects.

To clarify the relationship of the nAChR subunits and the neonicotinoids tested with the affinity and efficacy of neonicotinoids for the 12 fruit fly nAChRs (Table 1), we quantitatively analysed the factors governing the variations in the agonist activity indices (**Table 2, and Table H in S1 Text** for parameter and data sets). The adjusted coefficients of Dα1 and Dα2 subunits for affinity were -0.306 and -0.754, respectively (**Table 2**), suggesting that both subunits reduced the neonicotinoid affinity, the Dα2 contribution being higher than the Dα1 contribution, while the coefficient of Dα3 was 0.524, indicating that the subunit enhanced affinity (**Table 2**). Also, the compound properties underpin the affinity (**Table 2**). It was impossible to elucidate the contribution of the Dβ1 subunit since it is common to all the nAChRs being an essential subunit. However, we showed previously that the R81T mutation in the Dβ1 subunit strikingly reduced the affinity and efficacy of the neonicotinoids [30], indicating its critical role in determining neonicotinoid action [16, 35, 36]. On the other hand, the $I_{max}$ relied mainly on the compound properties even though the values also varied with subunit composition (**Fig 2C and Table 2**). The highest efficacy of clothianidin probably results from hydrogen bond formation of NH of its guanidine moiety with the backbone carbonyl of the tryptophan in loop B conserved in the insect α subunits [37].

The subunit factors governing variations in neonicotinoid affinity for the various fruit fly nAChRs were derived solely from the multivariate analyses. Therefore, to confirm the results,

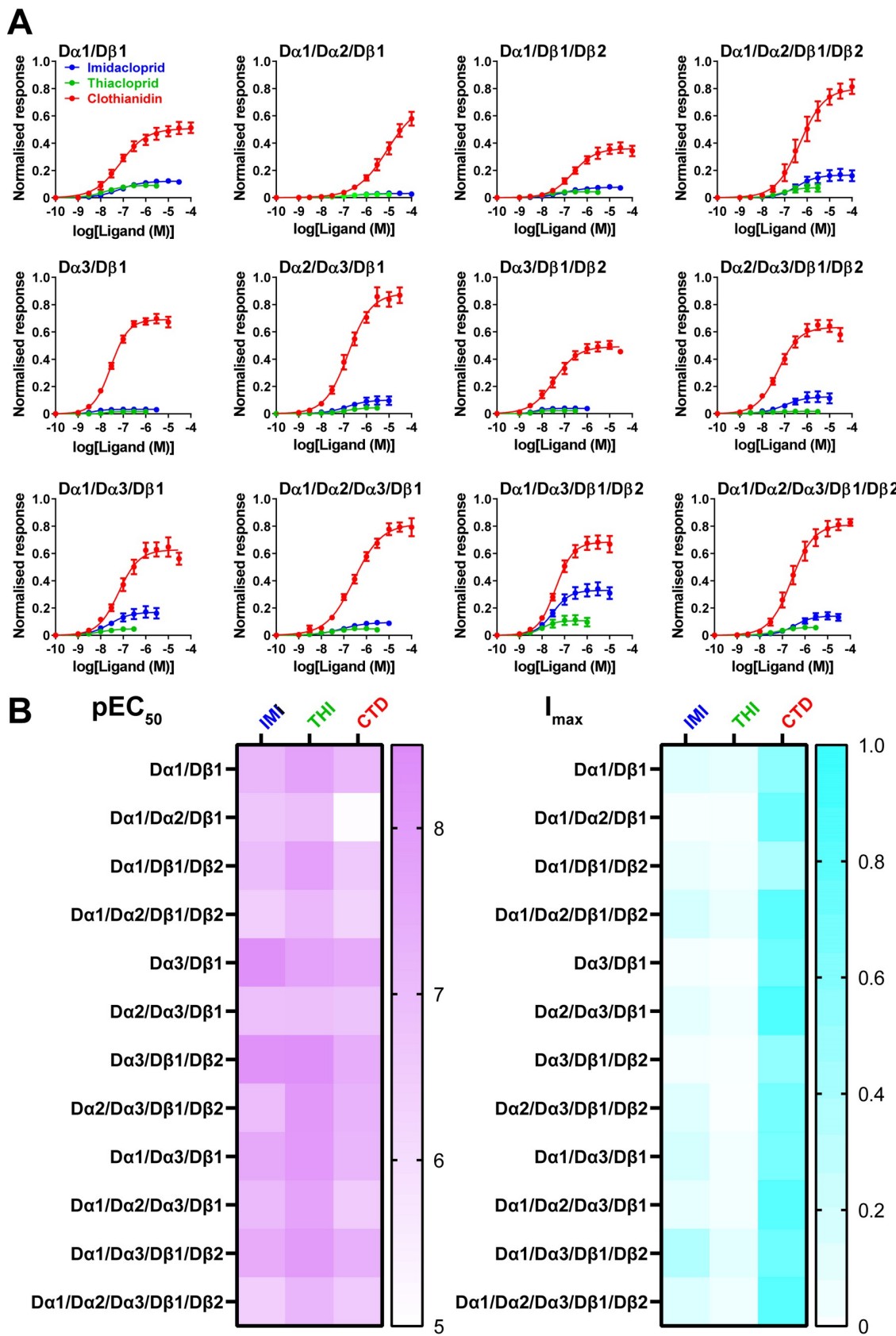

**Fig 3. Concentration-response relationships for agonist activity of imidacloprid, thiacloprid, and clothianidin for *D. melanogaster* nAChRs expressed in *X. laevis* oocytes and heatmap representations of the affinity and efficacy of neonicotinoids.** (**A**) Concentration-response relationships of agonist activity of neonicotinoids for *D. melanogaster* nAChRs. Each data plot represents mean ± standard error (n = 5). (**B**) Heatmap representation of the affinity in terms of $pEC_{50}$ values for the neonicotinoids tested. (**C**) Heatmap representation of the efficacy in terms of $I_{max}$ values for the neonicotinoids tested. See **Table 2** for the results of multivariate analyses of subunit and ligand factors governing variations in $pEC_{50}$ and $I_{max}$ values.

we performed the chaid (**Fig 4A**) and lattice (**Fig 4B**) analyses of the affinity of the neonicotinoids. In the chaid analysis, the Dα1 and Dα2 subunits were negative determinants, whereas the Dα3 subunit was a positive determinant of the affinity, Dα2 being a higher contributor than Dα1 and Dα3 (**Fig 4A**). Mean $pEC_{50}$ of all the neonicotinoids tested for nAChRs without Dα1 and Dα2, but with Dα3 was highest (7.506), indicating that Dα3 is the most critical determinant of high sensitivity for all the neonicotinoids tested of the *D. melanogaster* nAChRs. In the lattice analysis (**Fig 4B**), Dα2 subunit was a significant negative factor for the affinity (**Table I in S1 Text**).

Based on these results, we knocked down *Dα1*, *Dα2*, and *Dα3* subunit genes by using a pan-neuronal *Gal4* driver (*elav-Gal4*) and quantified the expression of genes encoding all nAChR subunits in Control (*elav-Gal4>w*$^{1118}$) and RNAi animals in both developmental (white pre-pupae) and adult stages of *D. melanogaster* (**Fig 5A**). At the same time, these RNAi animals were used to examine toxicity of imidacloprid, thiacloprid, and clothianidin (**Fig 5B**). We here focused on Dα1, Dα2, and Dα3 subunits, because these three subunits play critical roles in determining the affinity of the neonicotinoids (**Tables 1 and 2**). Knockdown of *Dα1*, *Dα2*, and *Dα3* differentially affected the other subunit gene expression level, depending on stage and sex (**Fig 5A**). During development, knockdown of each of *Dα1*, *Dα2*, and *Dα3* hardly affected other subunit gene expression except for *Dα2* RNAi, which significantly reduced *Dα1* expression. By contrast, knockdown of *Dα1* enhanced *Dβ3* expression in both males and females and *Dα7* expression in adult females. Knockdown of *Dα2* reduced *Dα1*, *Dα6,* and *Dα7* expression in both males and females, and *Dβ1* expression in adult males. Furthermore, knockdown of *Dα2* enhanced *Dβ3* expression in adult males. On the other hand, knockdown of *Dα3* reduced *Dα1* expression and enhanced *Dα2* expression in both males and females while enhancing

**Table 2. Multivariate analysis of factors determining the affinity ($pEC_{50}$) and efficacy ($I_{max}$) of neonicotinoids.**

|  | $pEC_{50}$ | $I_{max}$ |
|---|---|---|
| **Factors** | **Coefficients (95% CI)**[a] | **Coefficients (95% CI)**[a] |
| **Subunits** | | |
| Dα1 | -0.306 (-0.567 − -0.045)* | 0.053 (-0.008 − 0.113) |
| Dα2 | -0.754 (-0.967 − -0.541)* | 0.046 (-0.003 − 0.096) |
| Dα3 | 0.524 (0.264 − 0.785)* | 0.059 (-0.002 − 0.119) |
| Dβ2 | 0.092 (-0.121 − 0.305) | -0.003 (-0.052 − 0.047) |
| **Compounds** | | |
| Acetylcholine | Reference | Reference |
| Imidacloprid | 1.414 (1.113 − 1.715)* | -0.333 (-0.403 − -0.263)* |
| Thiacloprid | 1.785 (1.484 − 2.086)* | -0.881 (-0.951 − -0.811)* |
| Clothianidin | 2.229 (1.928 − 2.530)* | -0.951 (-1.020 − -0.881)* |

Data number and degree of freedom were 48 and 40, respectively. $R^2$ = 0.878 and 0.957 for $pEC_{50}$ and $I_{max}$, respectively. Akaike's information criterion was 46.7 and -93.5 for $pEC_{50}$ and $I_{max}$, respectively. CI, confidence interval. Coefficients for Dβ1 subunit could not be calculated because it was essential for the functional expression of the nAChRs compared.

[a]Coefficients were obtained by adjusted models using all covariates.

*Significant by *t*-test ($P < 0.05$).

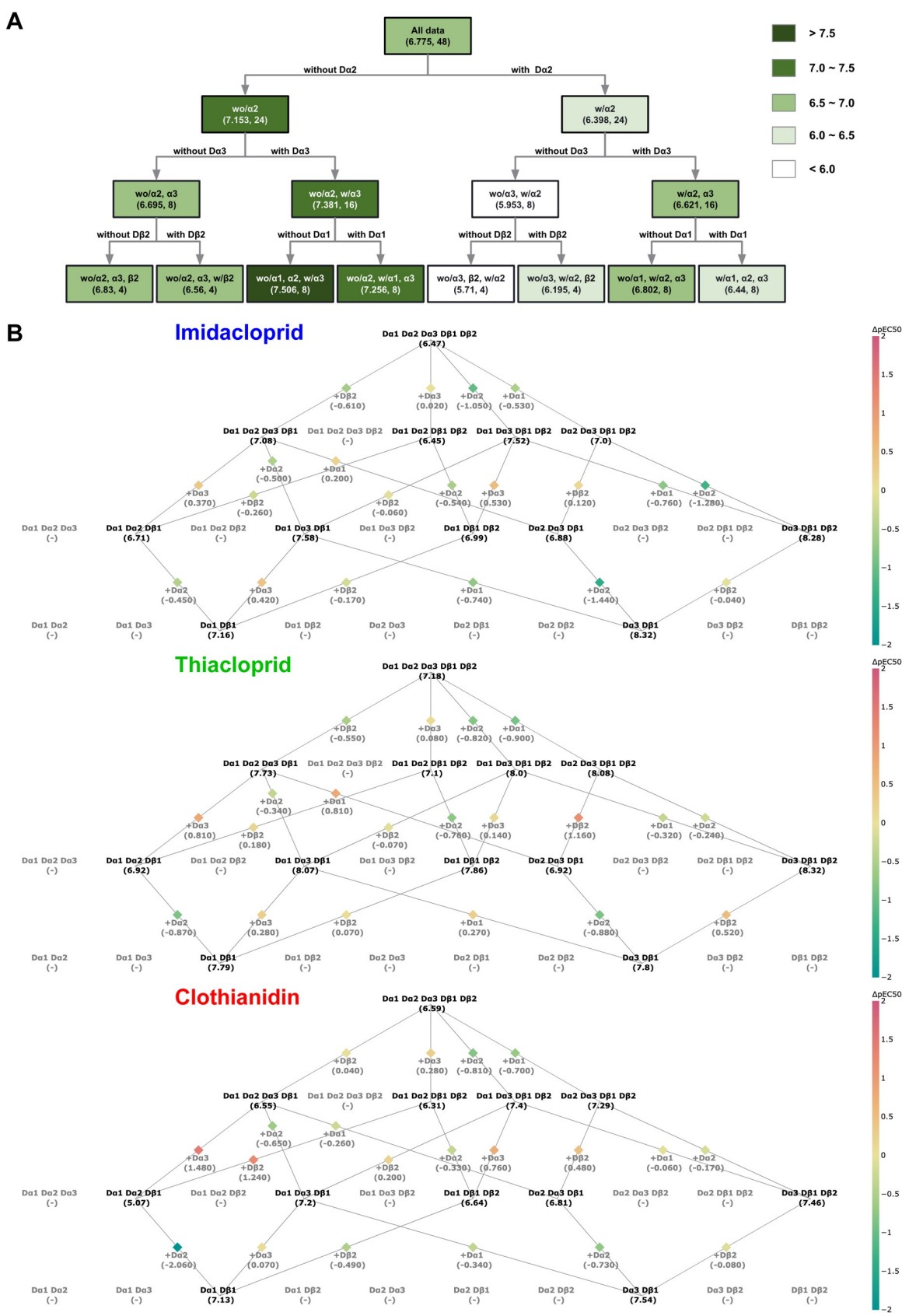

**Fig 4. Chaid and lattice analyses of subunit factors determining the affinity of neonicoticotinoid for the *D. melanogaster* nAChRs.** (A) Chaid analysis of subunit factors determining the affinity of neonicotinoids for the nAChRs. pEC$_{50}$ and data numbers are shown in each bracket. Abbreviations: with, w; without, wo. (B) Lattice analysis of subunit factors determining the neonicotinoid affinity for the nAChRs.

*Dβ3* expression in adult females (**Fig 5A**). These findings indicate that the subunit compensation occurs more frequently in adults than during development. Such subunit compensation can also enhance the inhibitory effect on the climbing behaviour, thereby inducing hypersensitisation to neonicotinoids.

Several studies investigated the effects of knocking out nAChR subunit genes in fruit flies on the toxicity of neonicotinoids to larvae or adults and showed that in almost all cases, such mutations resulted in reduced sensitivity to neonicotinoids [38–40].

Interestingly, Perry et al. showed that *Dα2* knockout enhanced nitenpyram sensitivity in larvae [38]. Also, Chen et al. showed that *Dα1/Dβ2* double knockouts reduced imidacloprid resistance level compared to that observed in a *Dα1* or a *Dα2* single gene knockout in *D. melanogaster* [41]. Still, the mechanism of these findings is not known. Liu et al. showed a higher Dα3 subunit contribution to the interactions with clothianidin than for other subunits tested in terms of lethal activity [40]. However, no such Dα3 preference for clothianidin was evident in our study. These findings may be attributable, at least in part, to the differences in the nAChR subunits involved in toxicity. Here we show that RNAi of *Dα1* and *Dα2* reduced toxicity of imidacloprid, thiacloprid, and clothianidin in larvae (**Fig 5B**), which is attributable to increased non-target/target nAChR ratio. However, as predicted by the multivariate analyses, RNAi of *Dα2* led to hypersensitivity to imidacloprid and thiacloprid in adult males and females and to clothianidin in adult males (**Fig 5B**), which counsels caution in believing that reduction of drug sensitivity generally happens in response to suppressing the primary target proteins. A direct interpretation of such an observation is that Dα2 subunit is the negative factor reducing the affinity of neonicotinoids (**Fig 4** and **Table 2**), hence reduced *Dα2* gene expression results in enhanced neonicotinoid sensitivity. The qRT-PCR data (**Fig 5A**) revealed that in response to RNAi of *Dα2*, expression of genes encoding Dα5, Dα6, and Dα7 subunits, of which the Dα5 and Dα6 subunits form low imidacloprid-sensitive nAChRs [42], was reduced, offering another explanation for the enhanced toxicity of neonicotinoids. Reduced toxicity by knockdown of *Dα2* and concomitant reduced *Dβ1* expression was also observed in adult males (**Fig 5A**), which can reduce numbers of nAChRs with neonicotinoid sensitivity since the Dβ1 subunit is essential for functional expression (**Fig 1** and **Table 1**). As such, subunit compensation in response to the knockdown of *Dα1*, *Dα2*, and *Dα3* varies with developmental stages and sexes as well as the primary target of RNAi, resulting in diverse neonicotinoid actions.

These data indicate that Dα1, Dα2, and Dα3 subunits all underpin the interactions with neonicotinoids in the fruit fly. Nevertheless, a contribution of the other subunits to the neonicotinoid actions should not be underestimated because subunit compensation, which can cause replacement of subunits in nAChRs, occurs in response to RNAi of each subunit gene (**Fig 5A**). The t-SNE representations of the single cell gene expression data indicate co-expression of the Dα1, Dα2, Dα3, Dβ1, and Dβ2 subunits with the Dα4 subunit (**Fig D in S1 Text**). The Dα1, Dα2, Dα3, Dβ1, and Dβ2 subunits also co-exist with Dβ3 subunit, although such cases are limited [43]. Hence, for the first time we evaluated the effects of replacing one of the Dα1, Dα2, and Dα3 subunits in the Dα1/Dα2/Dα3/Dβ1/Dβ2 nAChRs by the Dα4 or Dβ3 subunit on the agonist activity of imidacloprid, thiacloprid, and clothianidin as well as ACh (**Fig 6**, Fig E in **S1 Text** and **Table 1**). Except for the substitution of the Dα3 subunit, such switching increased affinity of neonicotinoids (**Fig 6, Table 1,** and **Table J in S1 Text**). For example, pEC$_{50}$ values of imidacloprid, thiacloprid, and clothianidin for the Dα1/Dα2/Dα3/β1/Dβ2

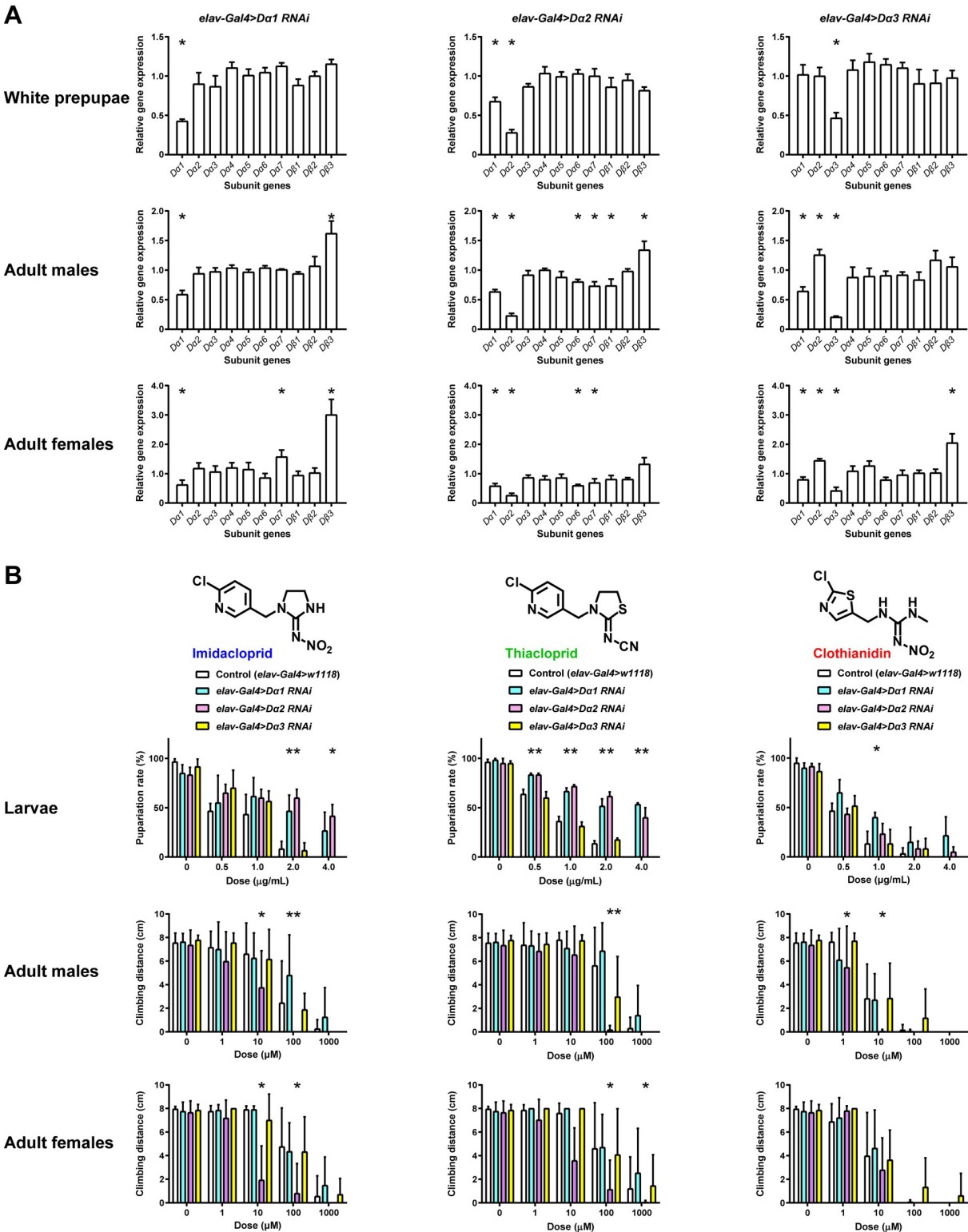

**Fig 5. Effects of α subunit gene RNAi on neonicotinoid toxicity in the fruit flies.** (**A**) Relative expression of genes encoding all nAChR subunits by pan-neuronal knockdown of *Dα1*, *Dα2*, and *Dα3* in white prepupae and adults of *D. melanogaster*. *elav-Gal4>UAS-dicer2* was used to induce RNAi in all neurons. Each bar indicates the mean ± standard deviation (n = 3). Asterisk indicates that the gene expression level changes as compared with the control (*elav-Gal4>w^{1118}*) (*t*-test, $P < 0.05$). (**B**) Toxicity of neonicotinoids by pan-neuronal knockdown of *Dα1*, *Dα2*, and *Dα3* in larvae and adults of *D. melanogaster*. Each bar represents the mean ± standard deviation (pupariation rate assay, n = 3;

adult climbing assay, control n = 20, RNAi n = 10). Asterisk indicates that the toxicity level changes compared with the control (one-way ANOVA, Bonferroni test, $P < 0.05$).

nAChR increased from 6.47, 7.18, and 6.59 to 7.40, 7.99, and 7.11, respectively by switching the Dα1 subunit to the Dα4 subunit. On the other hand, switching the Dα1 subunit of the Dα1/Dα2/Dα3/Dβ1/Dβ2 nAChR to the Dα4 or Dβ3 subunit reduced the efficacy of imidacloprid (Dα1/Dα2/Dα3/Dβ1/Dβ2 nAChR, 0.142, Dα2/Dα3/Dα4/Dβ1/Dβ2 nAChR, 0.027; Dα2/Dα3/Dβ1/Dβ2/Dβ3 nAChR, 0.021) and thiacloprid (Dα1/Dα2/Dα3/Dβ1/Dβ2 nAChR, 0.059, Dα2/Dα3/Dα4/Dβ1/Dβ2 nAChR, 0.018; Dα2/Dα3/Dβ1/Dβ2/Dβ3 nAChR, 0.010). Similarly, replacing the Dα2 subunit of the Dα1/Dα2/Dα3/Dβ1/Dβ2 nAChR with the Dα4 subunit reduced the efficacy of imidacloprid (0.142 to 0.101) and clothianidin (0.812 to 0.425), impairing fruit fly motility in accordance with the enhanced neonicotinoid toxicity by RNAi of *Dα2* (Fig 5B). By contrast, switching the Dα3 subunit to the Dα4 subunit had a minimal impact on the efficacy of imidacloprid and thiacloprid, while increasing that of clothianidin (0.812 to 1.001). Furthermore, switching the Dα3 subunit to the Dβ3 subunit reduced the efficacy of imidacloprid (0.142 to 0.080) and thiacloprid (0.059 to 0.023) while increasing that of clothianidin (0.812 to 0.974, Fig 6, Table 1 and Table J in S1 Text), explaining why targeting *Dα3* had less impact on neonicotinoid toxicity than targeting *Dα1* and *Dα2*. These results suggest that both Dα4 and Dβ3 subunits can form heteromeric nAChRs with either of Dα2/Dα3/Dβ1/Dβ2, Dα1/Dα3/Dβ1/Dβ2, and Dα1/Dα2/Dβ1/Dβ2 combinations and show unique pharmacological features in neonicotinoid actions. Also, it is conceivable that the nAChRs containing the Dα4 or Dβ3 subunit also contribute to the change of neonicotinoid toxicity in response to RNAi of *Dα1*, *Dα2*, and *Dα3*.

In conclusion, by studying 18 subunit combinations of subunits Dα1, Dα2, Dα3, Dα4, Dβ1, Dβ2, and Dβ3, we have found that imidacloprid, thiacloprid and clothianidin can interact with a broad range of *D. melanogaster* nAChRs formed not only by the Dα1, Dα2, Dα3, Dβ1, and Dβ2 subunits, but also by the Dα4 and Dβ3 subunits, which has not been described to the best of our knowledge. Although co-expression of these subunits does not necessary prove that they co-assemble to form functional nAChRs in neurons, it is clear that the three neonicotinoids exhibited diverse agonist actions on the 18 nAChRs tested, the outcome depending on both the compound as well as subunit composition. Notably, the Dα1, Dα2, Dα3, Dβ1, and Dβ2 subunits co-localise in organs underlying mating and egg laying, predicting that modulation of the nAChRs consisting of these subunits will affect the number of offspring. In future, it will be of considerable interest to test this hypothesis. If such actions are confirmed, not only for the fruit flies, but also for other insect species such as pollinators and disease vectors, this will counsel further caution in identifying target receptor subtypes simply in terms of reduced neonicotinoid sensitivity resulting only from gene disruption or suppression experiments.

## Methods

### Ethics statement

Oocytes at stage V or VI of development were removed under anesthetic (0.3 g L$^{-1}$ benzocaine) from adult female *X. laevis* according to the U.K. Animals (Scientific Procedures) Act, 1986. Care was always taken to minimise the number of animals used in experiments.

### ACh and neonicotinoids

ACh (#A6625) was purchased from MilliporeSigma (USA). The neonicotinoids (imidacloprid, #099–03771; thiacloprid, #205–19081; clothianidin, #034–22581) were purchased from FUJIFILM Wako Pure Chemical (Japan). These reagents were used without further purification.

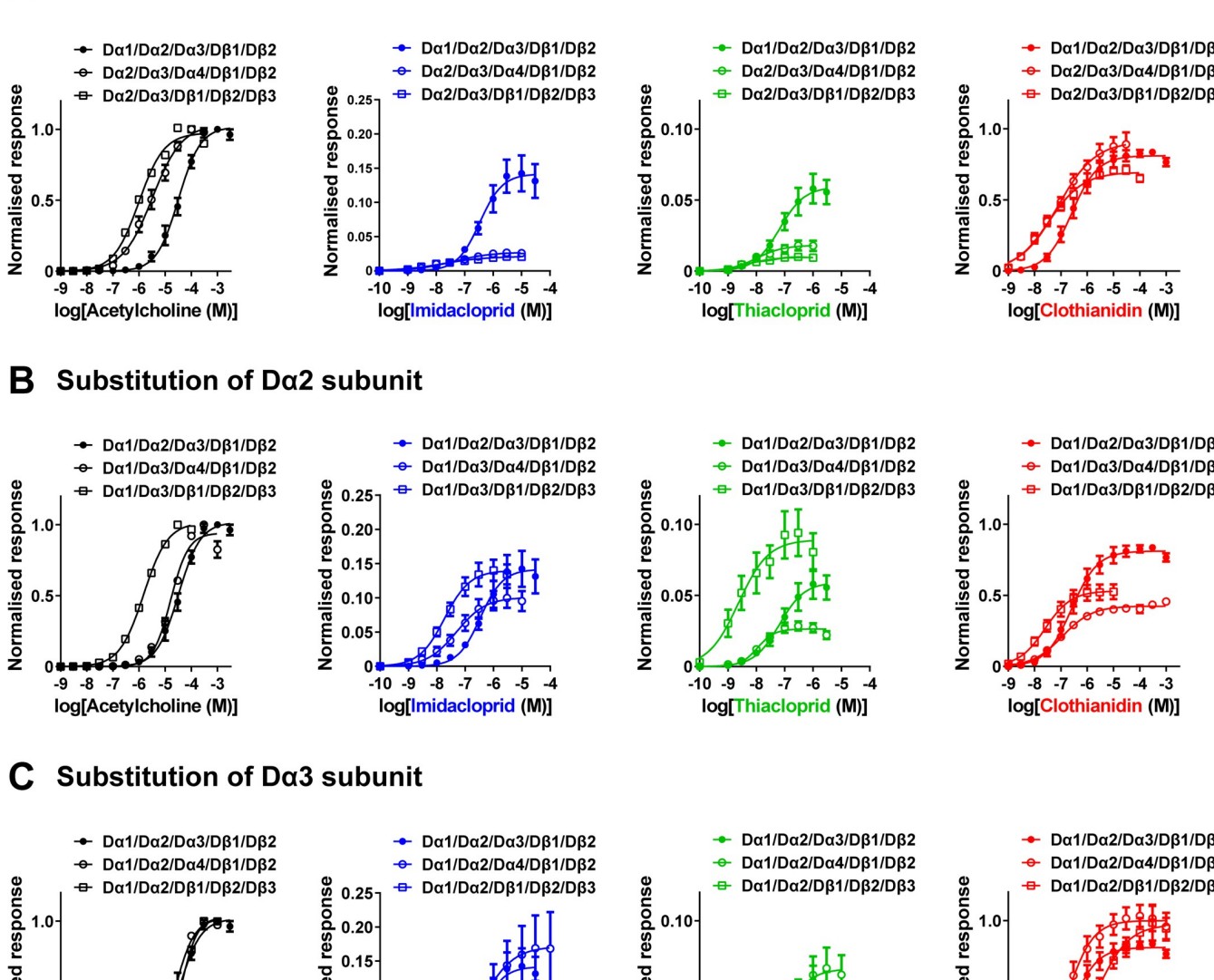

**Fig 6. Effects of Dα1, Dα2, and Dα3 subunit substitution by Dα4 or Dβ3 subunit on agonist actions of ligands on Dα1/Dα2/Dα3/Dβ1/Dβ2 *D. melanogaster* nAChRs.** Dα1 (**A**), Dα2 (**B**), and Dα3 (**C**) subunits were substituted by Dα4 or Dβ3 subunits and we compared the concentration-response curves for ACh, imidacloprid, thiacloprid, and clothianidin on the Dα1/Dα2/Dα3/Dβ1/Dβ2 nAChRs with those on the nAChRs resulting from the subunit switching. Except for the substitution of the Dα3 subunit, the subunit switching resulted in enhanced affinity of neonicotinoids. In the Dα1 and Dα2 subunit switching, Dα4 subunit addition resulted in reduced efficacy and enhanced affinity of imidacloprid and thiacloprid, whereas Dβ3 subunit addition increased the efficacy of thiacloprid. For the actions of clothianidin, Dα2 subunit switching by Dα4 or Dβ3 subunit reduced efficacy, while Dα3 subunit switching enhanced it. Each data point represents the mean ± standard error (n = 5).

## Flies

All flies were raised at 25˚C under 12 h/12 h light/dark cycle. The animals were reared on standard fly food containing 5.5 g agar, 100 g glucose, 40 g dry yeast, 90 g cornflour, 3 mL propionic acid, and 3.5 mL 10% butyl *p*-hydroxybenzoate (Nacalai Tesque, Japan, #06327–15) in

70% ethanol per liter. The control strain was $w^{1118}$, and transgenic flies are as follows: *UAS-nAChRα1 RNAi* (#28688) was obtained from the Bloomington Drosophila Stock Center (BDSC); *UAS-nAChRα2 RNAi* (*UAS-Dα2 RNAi*, #10760), *UAS-nAChRα3 RNAi* (*UAS-Dα3 RNAi*, #101806); *UAS-dicer2* (#60009) were obtained from the Vienna Drosophila Resource Center (VDRC); and *UAS-mCD8::GFP* (#108068) [44] was obtained from Kyoto Stock Center. *elav-Gal4 (3A3)* was obtained from Michael B. O'Connor. *nAChRα3*-knock-in *2A-Gal4* (*Dα3*-knock-in T2*A-Gal4*) was generated by the CRISPR/Cas9 system as described in detail below.

## Generation of *Dα3*-knock-in *T2A-Gal4* strain

The *Gal4* knock-in *D. melanogaster* flies were generated by CRISPR/Cas9-mediated homologous recombination. A targeting vector was designed such that the *T2A-Gal4* [45] is inserted in frame with the last intracellular region of the protein. The targeting vector and a gRNA expression vector that cuts near the target site were co-injected into fertilised eggs maternally expressing Cas9 protein. The flanking sequences of the insertion are: 5´-GAAAGAGGACTG GAAGTACGTGGCCATG/GTGCTCGATCGCCTGTTCCTGTGGATCTTCACAATAGC-3´ (The site of integration is indicated by a slash, The 20-bp gene-specific sequence of the gRNA is underlined.)

## Immunostaining

*D. melanogaster* male reproductive systems were dissected in Grace's Insect Medium, supplemented (Thermo Fisher Scientific, USA, #11605094) and fixed in 4% paraformaldehyde in Grace's medium for 30–60 min at room temperature (RT). The fixed samples were washed three times in phosphate-buffered saline (PBS) supplemented with 0.1% Triton X-100 (Nacalai Tesque, #12967–45). After washing, samples were blocked in the blocking solution (PBS with 0.1% Triton X-100 and 2% bovine serum albumin (MilliporeSigma, #A9647) for 1 h at RT then incubated with a primary antibody in the blocking solution at 4°C overnight. The primary antibodies used in this study were mouse anti-GFP monoclonal antibody (clone GFP-20; MilliporeSigma G6539; 1:1000) and rabbit anti-Tdc2 antibody (Abcam ab128225; 1:1000) [46]. Fluorophore (Alexa Fluor 488, or 546)-conjugated secondary antibodies (Thermo Fisher Scientific, #A11001, #A32732) were used at a 1:200 dilution and incubated for 2 h at RT in the blocking solution. After washing, all samples were mounted in FluorSave reagent (MilliporeSigma, #345789). Samples were visualised on an LSM 700 confocal microscope (Zeiss, Germany). Images were processed using the ImageJ package [47].

## cDNAs and cRNAs

cDNAs of the nAChR subunits and co-factors were cloned into pcDNA3.1 (+) vector (Thermo Fisher Scientific). The accession numbers of the nAChR subunits and cofactors are as follows: Dα1 (NP_524481), Dα2 (NP_524482), Dα3 (NP_525079), Dα4 (CAB77445), Dβ1 (NP_523927), Dβ2 (NP_524483), Dβ3 (NP_525098), DmRIC-3 (CAP16647), DmUNC-50 (NP_649813), and DmTMX3 (NP_648847). cRNAs were prepared using the mMESSAGE mMACHINE T7 ULTRA Transcription Kit (Thermo Fisher Scientific, #AM1345) according to the manual with the cDNA template which was cut with appropriate restriction enzymes at the 3' end of the cDNA.

## cRNA expression in *X. laevis* oocytes

We minimised the use of *X. laevis* according to the UK Animals (Scientific Procedures) Act, 1986. Female *X. laevis* were anaesthetised with benzocaine (Nacalai Tesque, #14804–92) prior to oocyte excision. Oocytes were defolliculated after collagenase treatment in $Ca^{2+}$-free

standard oocyte saline ($Ca^{2+}$-free SOS). cRNAs of the nAChR subunits and co-factors mixed at a concentration of 0.1 mg/mL was injected into oocytes at a volume of 50 nL. Then the oocytes were incubated in the incubation medium (SOS supplemented with sodium pyruvate (Nacalai Tesque, #13058–12), penicillin-streptomycin (MilliporeSigma, #P4333), gentamycin (Nacalai Tesque, #16672–04), and 4% horse serum (Thermo Fisher Scientific, #26050–070) for 3–4 days prior to electrophysiology [30].

## Voltage-clamp electrophysiology

Each defolliculated *X. laevis* oocyte was secured in a Perspex recording chamber and perfused with the standard oocyte saline (SOS) containing 0.5 µM atropine (SOSA) at a flow rate of 7–10 mL/min [48]. Two glass electrodes filled with 2 M KCl were impaled into each oocyte and the membrane potential was clamped at -100 mV. ACh and α-BTX (Alamone Labs, Israel, #B-100) were dissolved directly in SOSA, while test solutions of the neonicotinoids were diluted to the final concentration from DMSO stock solutions. DMSO at 1% (v/v) or lower had no effect on the responses to neonicotinoids or other ligands tested. ACh and neonicotinoids were applied for 5 s successively at 3 min intervals. α-BTX was tested as previously described in the literature [33]. The peak amplitude of the response was measured and analysed by pCLAMP (Molecular Devices, USA). The agonist response data were normalised to the maximum response to ACh at concentrations at which the response amplitude attained plateau and fitted by non-linear regression using Prism (GraphPad Software, USA), according to the following equation.

$$Y = \frac{I_{max}}{1 + 10^{(logEC_{50} - X)n_H}}$$

Where Y is the normalised response, X is log[ligand (M)], $EC_{50}$ is the half maximal concentration (M), $I_{max}$ is the maximum normalised response and $n_H$ is the Hill coefficient.

## Total RNA extraction and quantitative reverse transcription (qRT)-PCR

Animals were collected in 1.5 ml tubes and immediately flash-frozen in liquid nitrogen. Total RNA from white prepupa (0 hour after puparium formation) or adults (3 days after eclosion) was extracted using RNAiso Plus (Takara Bio, Japan, #9109) according to the manufacturer's instructions. cDNA was generated from purified total RNA using ReverTra Ace qPCR RT Master Mix with gDNA Remover (TOYOBO, Japan, #FSQ-301). qRT-PCR was performed on the Thermal Cycler Dice TP800 system (Takara Bio) using THUNDERBIRD Next SYBR qPCR Mix (Takara Bio, #QPX-201). For absolute quantification of mRNAs, serial dilutions of plasmids containing coding sequences of the target genes or *rp49* were used for standards. After the molar amounts were calculated, transcript levels of the target mRNA were normalised to *rp49* levels in the same samples. The primers used are listed in **Table K in S1 Text.** The primers to detect *rp49* levels are as previously reported [49].

## Pupariation rate assay

Eggs were laid on grape juice plates with yeast paste at 25˚C for 6 h. After 24 h, early first instar larvae just after hatching (20 larvae/vial) were transferred into a mini-vial (Sarstedt, Germany, #58.487) with 2.0 g of neonicotinoid feeding assay food: 50 mL eq. of blue food powder (Formula 4–24 Instant Drosophila Medium, Carolina, USA, #173210), 50 mL eq. of yeast powder (Brewer's yeast, MP Biomedicals, USA, #903312), and 100 mL $dH_2O$ containing 0.1% dimethyl sulfoxide (DMSO; Nacalai Tesque, #13407–45) (for control) or one of the neonicotinoids (imidacloprid, thiacloprid, and clothianidin) in 0.1% DMSO. After a week of incubation at 25˚C, pupal numbers were scored in each vial.

## Adult climbing assay

Adult *D. melanogaster* flies were collected within a day following eclosion and placed in normal fly food (less than 30 flies/vial). Flies were transferred daily to new fly food. 2–5 days after eclosion, flies were briefly anesthetised with $CO_2$, and the sexes were separated and sorted into fly vials containing 1.0% agar food for starvation (10 flies/vial). After 16 h starvation, flies were transferred to vials containing neonicotinoid-containing food without anesthesia and cultured for 6 h (10 flies/vial). Neonicotinoid-containing foods were prepared by mixing 10 μL of diluted neonicotinoids dissolved in DMSO with 990 μL of a solution containing 1% agar and 5% sucrose for each vial. After 6 h cultured in vials containing neonicotinoid-containing food, flies were gently tapped down to the surface of the food, and flies that climbed within 20 s after tapping were recorded by a video camera (GZ-F270-W, JVCKENWOOD, Japan). The maximum climbing heights of the flies within 20 s after tapping were measured using ImageJ1.53v (National Institute of Health, USA). Since the height from the surface of the food to the vial top is 8 cm, the maximum climbing height is 8 cm.

## Reproducibility of data

At least two authors participated independently in measuring data to confirm reproducibility of the results. For electrophysiology, five oocytes from at least two frogs were used to determine the agonist activity of each ligand at each concentration.

## Statistical analyses

Prism software was employed for the statistical analyses. The peak current amplitude of the agonist actions of ACh at 100 μM was compared between the nAChRs by Kruskal-Wallis test. One-way ANOVA was used to analyse the differences of the ligand agonist activity in terms of $pEC_{50}$ on the various nAChRs expressed in *X. laevis* as well as data obtained with *D. melanogaster* larvae and adults.

## Multiple variate analysis

The multiple variate analysis was conducted with python to examine if *D. melanogaster* nAChR subunits and compounds contribute significantly to the agonist activity in terms of $pEC_{50}$ and $I_{max}$. We used a dataset including 48 samples (**Table H in S1 Text**). Objective variables are $pEC_{50}$ and Imax and explanatory variables are subunits (Dα1, Dα2, Dα3, and Dβ2) and compounds (ACh, imidacloprid, thiacloprid, and clothianidin). Data for ACh were used as references when calculating the subunit and compound factors governing the variations in the agonist activity indices.

## Chaid analysis

Chaid analysis was conducted with python to examine if the nAChR subunits contribute significantly to the agonist activity in $pEC_{50}$. Parameter max depth was set as 4. Objective variable is $pEC_{50}$ and explanatory variables are subunits (Dα1, Dα2, Dα3, and Dβ2) and compounds (ACh, imidacloprid, thiacloprid, and clothianidin).

## Lattice visualization

The lattice visualization was used to observe the positive contribution to $pEC_{50}$ of adding each subunit (Dα1, Dα2, Dα3, Dβ1, and Dβ2). The presence or absence of subunits forms the powerset with lattice structure with respect to the inclusion order. The data are grouped by differences between two sets of subunits and denoted by "+<subunit name>" on each edge. The color bar indicates $\Delta pEC_{50}$, the value obtained by subtracting $pEC_{50}$ for smaller nAChR

subunit set from that for larger nAChR subunit set. The significance of the $\Delta pEC_{50}$ values was analysed by the 95% confidence interval.

Data used in Figs 1–3, 5, and 6 are available from Dryad [50].

## Dryad DOI

https://doi.org/10.5061/dryad.qz612jmk5 [50]

## Supporting information

**S1 Text. Supporting information. Fig A.** Inward current response of oocytes expressing *D. melanogaster* nAChRs to several concentrations of ACh. Horizontal bar shows application of ACh. **Fig B.** Effects of α-bungarotoxin (α-BTX) on the response to 100 μM ACh of *X. laevis* oocytes expressing *D. melanogaster* nAChRs. (**a**) Inward currents induced in oocytes expressing *D. melanogaster* nAChRs in response to 100 μM ACh in the absence and presence of α-BTX. Horizontal lines indicate application of ACh. (**b**) Bar graph representations of the current amplitude of responses to 100 μM ACh of the nAChR expressing oocytes exposed to 10 nM or 100 nM α-BTX. Error bars are standard error of the mean (n = 5). **Fig C.** Inward current responses to ACh and neonicotinoids (imidacloprid, thiacloprid, and clothianidin) of oocytes expressing *D. melanogaster* nAChRs. Horizontal lines indicate application of neonicotinoids. **Fig D.** t-SNE representations of *Dα1*, *Dα2*, *Dα3*, *Dα4*, *Dβ1*, and *Dβ2* gene expressions in the adult brain and ventral nerve cord of *D. melganogaster*. The figure was illustrated by SCope (https://scope.aertslab.org/#/86757313-d473-4f5f-b045-fc035d99451a/*/welcome) using single cell RNA-sequencing data [43]. These six nAChR subunit genes are co-expressed in single cells (See white dots.). **Fig E.** Inward current responses to ACh and neonicotinoids (imidacloprid, thiacloprid, and clothianidin) of oocytes expressing *D. melanogaster* Dα2/Dα3/Dα4/Dβ1/Dβ2, Dα2/Dα3/Dβ1/Dβ2/Dβ3, Dα1/Dα3/Dα4/Dβ1/Dβ2, Dα1/Dα3/Dβ1/Dβ2/Dβ3, Dα1/Dα2/Dα4/Dβ1/Dβ2, and Dα1/Dα2/Dβ1/Dβ2/Dβ3 nAChRs. Horizontal lines indicate application of neonicotinoids. **Table A.** One-way ANOVA of the $pEC_{50}$ values of ACh for *D. melanogaster* nAChRs. **Table B.** One-way ANOVA of the $pEC_{50}$ values of imidacloprid for *D. melanogaster* nAChRs. **Table C.** One-way ANOVA of the $I_{max}$ values of imidacloprid for *D. melanogaster* nAChRs. **Table D.** One-way ANOVA of the $pEC_{50}$ values of thiacloprid for *D. melanogaster* nAChRs. **Table E.** One-way ANOVA of the $I_{max}$ values of thiacloprid for *D. melanogaster* nAChRs. **Table F.** One-way ANOVA of the $pEC_{50}$ values of clothianidin for *D. melanogaster* nAChRs. **Table G.** One-way ANOVA of the $I_{max}$ values of clothianidin for *D. melanogaster* nAChRs. **Table H.** Data set for multivariate analyses. **Table I.** Mean and 95% confidence intervals of $\Delta pEC_{50}$ values obtained by lattice analysis. **Table J.** One-way ANOVA of the $pEC_{50}$ and $I_{max}$ values of ligands for *D. melanogaster* nAChRs containing Dα4 or Dβ3 subunit as compared with the values for Dα1/Dα2/Dα3/Dβ1/Dβ2 nAChR. **Table K.** Primers for qRT-PCR
(DOCX)

## Acknowledgments

We acknowledge KYOTO Stock Center (DGRC) in Kyoto Institute of Technology for supplying the fruit fly strain.

## Author Contributions

**Conceptualization:** Yuma Komori, Koichi Takayama, Naoki Okamoto, Makoto Ihara, Ryusuke Niwa, David B. Sattelle, Kazuhiko Matsuda.

**Data curation:** Makoto Ihara, Ryusuke Niwa, Kazuhiko Matsuda.

**Formal analysis:** Yuma Komori, Koichi Takayama, Naoki Okamoto, Makoto Ihara, Daitaro Misawa, Kotaro Kamiya, Kazuki Seike, Ryusuke Niwa, Kazuhiko Matsuda.

**Funding acquisition:** Makoto Ihara, Ryusuke Niwa, Kazuhiko Matsuda.

**Investigation:** Yuma Komori, Koichi Takayama, Naoki Okamoto, Masaki Kamiya, Wataru Koizumi, Makoto Ihara, Daitaro Misawa, Kotaro Kamiya, Yuto Yoshinari, Kazuki Seike, Ryusuke Niwa, Kazuhiko Matsuda.

**Methodology:** Daitaro Misawa, Kotaro Kamiya, Shu Kondo, Hiromu Tanimoto, Ryusuke Niwa, Kazuhiko Matsuda.

**Project administration:** Ryusuke Niwa, Kazuhiko Matsuda.

**Resources:** Shu Kondo, Hiromu Tanimoto, Ryusuke Niwa, David B. Sattelle, Kazuhiko Matsuda.

**Supervision:** Ryusuke Niwa, Kazuhiko Matsuda.

**Validation:** Naoki Okamoto, Makoto Ihara, Ryusuke Niwa, David B. Sattelle, Kazuhiko Matsuda.

**Visualization:** Yuma Komori, Naoki Okamoto, Daitaro Misawa, Kotaro Kamiya, Yuto Yoshinari, Kazuki Seike, Ryusuke Niwa, Kazuhiko Matsuda.

**Writing – original draft:** Yuma Komori, Koichi Takayama, Naoki Okamoto, Makoto Ihara, Daitaro Misawa, Kotaro Kamiya, Kazuki Seike, Ryusuke Niwa, David B. Sattelle, Kazuhiko Matsuda.

**Writing – review & editing:** Yuma Komori, Koichi Takayama, Naoki Okamoto, Makoto Ihara, Daitaro Misawa, Kotaro Kamiya, Ryusuke Niwa, David B. Sattelle, Kazuhiko Matsuda.

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
