## [Decision Letter · Decision Letter 0]

31 Oct 2022

Dear Dr Matsuda,

Thank you very much for submitting your Research Article entitled 'Functional impact of subunit composition and compensation on Drosophila melanogaster nicotinic receptors – targets of neonicotinoids' to PLOS Genetics.

The manuscript was fully evaluated at the editorial level and by independent peer reviewers. The reviewers appreciated the attention to an important topic but identified some concerns that we ask you address in a revised manuscript.

We therefore ask you to modify the manuscript according to the review recommendations. Your revisions should address the specific points made by each reviewer.

Yours sincerely,

Subba Reddy Palli, Ph.D.

Academic Editor

PLOS Genetics

Gregory P. Copenhaver

Editor-in-Chief

PLOS Genetics

Reviewer's Responses to Questions

**Comments to the Authors:**

Reviewer #1: The manuscript by Komori provides exciting new data regarding the heterologous expression of insect nicotinic acetylcholine receptors (nAChRs), enhancing our understanding of different receptor subtypes, their sensitivity to neonicotinoid insecticides and redundancy/compensation of various subunits. A lot of data has been provided between the main manuscript and the supplementary information, which will provide a wealth of information to those studying insect nAChRs.

I have only a few minor suggestions (page numbers refer to the pdf):

1/ Abstract second line: "strongly concerned." should be reworded, perhaps to "of serious concern."

2/ Abstract: Bombus terrestris (spelling)

3/ Abstract: "The presence of Da1 and Da2 subunits reduced the affinity of nAChRs for imidacloprid.....". How did you measure this? Mention heterologous expression of subunits in Xenopus laevis oocytes.

4/ p9: As with the current amplitude, give a specific example of how Da3 significantly changed the EC50.

5/ p14 fifth line: "knockdown of Da1 or Da2", otherwise it reads like you knocked down both Da1 and Da2.

6/ p15: provide a couple of examples of how the addition of Da4 or Db3 to, for example, Da2/Da3/Db1/Db2, resulted in a significant change of EC50 and max current.

7/ p20: It appears that relative levels of cDNA, compared to the control strain, were measured by qPCR as opposed to quantifying absolute amounts of RNA present. Was the deltaCt method used?

8/ p20: "Larval feeding assay" sounds like you are trying to quantify how much feeding is taking place where really it appears that you are scoring pupariation. Perhaps you should change the subheading to reflect this.

9/ A paper this year, Chen et al. DOI: 10.1016/j.pestbp.2022.105118, uses dual knockout of nAChRs in Drosophila. This paper should be considered and discussed in light of findings in the current study.

Reviewer #2: This manuscript reports functional and pharmacological characterization of nicotinic acetylcholine receptor (nAChRs) subunits from Drosophila melanogaster. The same group successfully expressed various combinations of nAChR subunits (Dα1, Dα2, Dβ1, and Dβ2) with several cofactors in Xenopus oocytes in an earlier study published in PNAS. In this study, they included Dα3 in various subunit combinations for functional expression in oocytes and examined the effect of three neonicotinoid insecticide, imidacloprid, thiacloprid, and clothianidin, on nAChRs in Xenopus oocytes. They concluded that Dα1, Dα2 and Dα3 subunits are likely determinants of insecticide toxicity. Then they carried out RNAi of these three subunits and insecticide bioassay using these mutants. They observed subunit compensation and enhanced toxicity in all three RNAi lines.

Major comments:

As indicated by the authors, several recent studies used CRISPR-Cas9 to knockout individual nAChR genes to evaluate the role of nAChRs in physiology and pharmacology. One of the studies by Lu et al. 2022, "Nicotinic acetylcholine receptor modulator insecticides act on diverse receptor subtypes with distinct subunit compositions | PLOS Genetics" reported the effects of individual knockout of ten nAChRs (seven α and three β) on Drosophila susceptibility to eleven nAChR-targeting insecticides including the three insecticides tested in this study. Knockout of Dα1 or Dα2, but not Dα3, conferred resistance to imidacloprid, thiacloprid or clothianidin. Also, none of the subunit compensation reported in this study was observed in that study.

Please compare results in Lu et al. (2022) with yours, including subunit compensation and toxicity data, and explain the discrepancy

Other comments:

Why were “pupariation rate” and “climbing distance”, but not mortality, used as endpoints for insecticide toxicity?

In the conclusion, the authors claim “Notably, the Dα1, Dα2, Dα3, Dβ1, and Dβ2 subunits co-localise in organs underlying mating and egg laying, predicting that modulation of the nAChRs

consisting of these subunits will affect the number of offspring” Since these subunits are expressed in neurons that are related to both female and male reproductive functions, the authors should conduct bioassays to evaluate egg laying using their RNAi lines. Is the Dα3 subunit expressed elsewhere in the nervous system besides ejaculatory duct neurons?

It is still unclear whether these combinations of nAChRs tested in Xenopus oocytes actually exist in vivo. Detection of coexpression of Dα3 subunit with Dα1, Dα2, Dβ1, and Dβ2 subunits in the same neurons does not necessarily reveal the subunit composition of native nAChR receptors in these neurons. The authors need to discuss the limitation of their approaches.

**Have all data underlying the figures and results presented in the manuscript been provided?**

Reviewer #1: Yes

Reviewer #2: Yes

PLOS authors have the option to publish the peer review history of their article (what does this mean?). If published, this will include your full peer review and any attached files.

Reviewer #1: No

Reviewer #2: No

---

## [Editor Report · Decision Letter 1]

11 Nov 2022

Dear Dr Matsuda,

We are pleased to inform you that your manuscript entitled "Functional impact of subunit composition and compensation on Drosophila melanogaster nicotinic receptors – targets of neonicotinoids" has been editorially accepted for publication in PLOS Genetics. Congratulations!

Yours sincerely,

Subba Reddy Palli, Ph.D.

Academic Editor

PLOS Genetics

Gregory P. Copenhaver

Editor-in-Chief

PLOS Genetics

Comments from the reviewers (if applicable):

**Data Deposition**

http://datadryad.org/submit?journalID=pgenetics&manu=PGENETICS-D-22-01133R1

**Press Queries**

---

## [Editor Report · Acceptance letter]

26 Jan 2023

PGENETICS-D-22-01133R1 

Functional impact of subunit composition and compensation on Drosophila melanogaster nicotinic receptors – targets of neonicotinoids 

Dear Dr Matsuda, 

We are pleased to inform you that your manuscript entitled "Functional impact of subunit composition and compensation on Drosophila melanogaster nicotinic receptors – targets of neonicotinoids" has been formally accepted for publication in PLOS Genetics! Your manuscript is now with our production department and you will be notified of the publication date in due course.

With kind regards,

Orsolya Voros

PLOS Genetics

On behalf of:
